# Biodiversity increases resistance of grasslands against plant invasions under multiple environmental changes

Cai Cheng [1,2], Zekang Liu [2], Wei Song[2], Xue Chen[2], Zhijie Zhang [3], Bo Li [4], Mark van Kleunen [3,5] & Jihua Wu [1] ✉

Biodiversity often helps communities resist invasion. However, it is unclear whether this diversity–invasion relationship holds true under environmental changes. Here, we conduct a meta-analysis of 1010 observations from 25 grassland studies in which plant species richness is manipulated together with one or more environmental change factors to test invasibility (measured by biomass or cover of invaders). We find that biodiversity increases resistance to invaders across various environmental conditions. However, the positive biodiversity effect on invasion resistance is strengthened under experimental warming, whereas it is weakened under experimentally imposed drought. When multiple factors are imposed simultaneously, the positive biodiversity effect is strengthened. Overall, we show that biodiversity helps grassland communities resist plant invasions under multiple environmental changes. Therefore, investment in the protection and restoration of native biodiversity is not only important for prevention of invasions under current conditions but also under continued global environmental change.

The Anthropocene has seen a rapid increase in invasions by alien species as well as by range-expanding native species[1–3]. Such invasions may pose a major threat to biodiversity, the economy, and human well-being[4,5]. There are many factors that affect the likelihood of species invasions, including background climatic conditions, the magnitude and type of anthropogenic environmental change, and biotic features of the community (e.g. the types and diversity of native species), all of which can interact[6,7]. Among the many hypotheses in invasion biology addressing these factors[8], much attention has been paid to the biotic resistance hypothesis[9,10], which predicts that more diverse communities should be more resistant to species invasions.

Empirical support for the biotic resistance hypothesis has been mixed. While some large-scale observational studies show compelling evidence for negative relationships between native diversity and invasion[7,11], these observational studies have limited ability to infer causality. This is because both native residents and invaders respond to variation in the environment and to each other[12,13]. Indeed, across larger spatial focal units (i.e. regions), there are often positive correlations between native and alien species richness because both groups of species respond in similar ways to the environmental conditions in the regions[14–16], even if there are negative relationships at smaller spatial scales[12,17]. Given the limited causal inference of observational studies, the most definitive way to examine the relationship between

[1]State Key Laboratory of Herbage Improvement and Grassland Agro-Ecosystems, College of Ecology, Lanzhou University, Lanzhou 730000, China. [2]Ministry of Education Key Laboratory for Biodiversity Science and Ecological Engineering, National Observations and Research Station of Wetland Ecosystems of the Yangtze Estuary, Institute of Biodiversity Science and Institute of Eco-Chongming, School of Life Sciences, Fudan University, Shanghai 200438, China. [3]Department of Biology, University of Konstanz, Konstanz 78464, Germany. [4]Ministry of Education Key Laboratory for Transboundary Ecosecurity of Southwest China, Yunnan Key Laboratory of Plant Reproductive Adaptation and Evolutionary Ecology and Centre for Invasion Biology, Institute of Biodiversity, School of Ecology and Environmental Science, Yunnan University, Kunming 650504, China. [5]Zhejiang Provincial Key Laboratory of Plant Evolutionary Ecology and Conservation, Taizhou University, Taizhou 318000, China. ✉e-mail: wjh@lzu.edu.cn

diversity and invasion is through experiments that manipulate the diversity (e.g. species richness) of the resident community and measure its resistance to invasion (e.g. biomass or cover of invaders)[18]. Indeed, many such experiments corroborate the positive relationship between diversity and invasion resistance[19–22]. However, there is considerable variability in the strength of the relationship[23,24], and a number of exceptions also occur[25,26]. Likely, this variation in the strength of the relationship between diversity and invasion resistance is caused by variations in environmental conditions[27–29].

Earth's ecosystems are exposed to numerous environmental change factors[30], such as climate change, eutrophication, overgrazing and pesticide use, all of which can have profound consequences for resident biota and invaders. According to the stress-gradient hypothesis, species interactions could switch from strong competition in favorable environments to weak competition or even facilitation in stressful environments[28,31]. Therefore, we might expect a stronger biodiversity effect on invasion resistance when the communities face stressful factors (i.e. impairing the overall performance of plants) as these should enhance positive interactions between native species. In contrast, the biodiversity effect would be weakened by favorable factors (i.e. benefiting the overall performance of plants). In addition to the effect of environmental change factors on interactions between native species, the relationship between diversity and invasion resistance could also be influenced by different responses of alien and native species to environmental change factors[32]. However, it remains unclear whether and how the different environmental change factors affect the relationship between diversity and invasion resistance.

In addition to the type of environmental change factors, the number of simultaneously acting factors may also influence the relationship between diversity and invasion resistance. While different factors can additively influence the relationship between diversity and invasion resistance[33], they could also act synergistically or antagonistically[34,35]. Even though the joint effects of multiple factors on either resident biota or invaders have been reported[36,37], we still lack information about how the relationship between diversity and invasion resistance responds to multiple simultaneously acting factors. This gap may result from the complex and large experimental designs that are needed when multiple levels of biodiversity are crossed with numerous environmental change factors. However, a recent study reported that increasing the number of simultaneously acting factors caused increasingly stressful environments[38], suggesting that there might be a stronger biodiversity effect on invasion resistance in the face of multiple simultaneous factors. Knowledge about the effect of multiple factors on the relationship between diversity and invasion resistance is necessary to boost our confidence that promoting native biodiversity in order to reduce invasions is a viable option under realistic global change scenarios.

Typically, biodiversity helps resist invasion primarily through enhanced competitive suppression (e.g. due to higher productivity) of resident species on invaders, mainly via complementarity and selection effects[20]. Complementarity effects occur when more diverse communities have more species that more fully occupy the available niche space, thereby pre-empting opportunities for invaders. Selection effects occur when more diverse communities have a higher probability of containing species that have greater competitive ability against invaders. As such, the biodiversity effect on invasion resistance will be influenced by multiple experimental factors. For example, a greater number of resident plant species, a longer duration of the experiment and smaller experimental units should result in stronger complementarity effects, and should reduce the niche space available for invaders[39]. Furthermore, the biodiversity effect on invasion resistance could also be influenced by the type of invaders. Several biodiversity experiments refer to invaders as any species that has not been planted in a given experimental unit[40]. Among these invaders, species that are residents in other experimental units (i.e. internal invaders)

should, due to a priority effect, be more likely to invade than novel external invaders −particular alien ones− that are not part of the experiment's resident species pool.

Here, we conduct a meta-analysis on 1010 observations from 25 grassland studies in which plant species richness is experimentally manipulated together with one or more environmental change factors. These factors include warming, drought, elevated atmospheric $CO_2$, eutrophication, pesticide use, grazing by domestic animals, human-caused fire, physical disturbance, and combinations of two or three of these factors. Our main objective is to assess whether and how the type and number of environmental change factors affect the biodiversity effect on invasion resistance. We measure invasion resistance of the resident community by the performance (biomass or percent cover) of all invaders of an experimental unit. However, when possible, we also distinguish for each experimental unit between internal invaders and external invaders. For the latter, we also distinguish between native and alien invaders (non-native to the experiment site). We here hypothesize that: (1) plant diversity increases the resistance of grasslands against invaders, with the strongest resistance to alien external invaders, (2) the biodiversity effect on invasion resistance is positively correlated with the effect on resident productivity and becomes stronger with increasing resident species richness and experimental duration, and with smaller sizes of the experimental units, (3) the biodiversity effect on invasion resistance is strengthened by stressful factors (e.g. drought, grazing and fire) but weakened by favorable factors (e.g. warming, elevated atmospheric $CO_2$ and eutrophication), and (4) the biodiversity effect on invasion resistance is strengthened by multiple simultaneous factors. By testing these hypotheses, our study provides evidence that plant diversity increases the resistance of grasslands against plant invasions. This is also the case under environmental changes, although the magnitude of the positive biodiversity effect increases or decreases, depending on the type and number of environmental change factors.

## Results

Averaged across all studies, we found a significantly positive effect of biodiversity on invasion resistance, both under ambient conditions and in the presence of environmental change factors (Fig. 1a; Supplementary Fig. 1a). The positive biodiversity effect on invasion resistance was strengthened by warming ($Q_M = 8.77$, $p = 0.003$) and weakened by drought ($Q_M = 7.06$, $p = 0.008$), but was not significantly affected by the other factors (Fig. 1a). This was also reflected by significant effect sizes of ΔNBE −difference in the net biodiversity effect (NBE) between manipulated and ambient conditions− under warming (mean = 0.82, 95% CI = [0.05, 1.59]) and drought (mean = −0.50, 95% CI = [−0.77, −0.22]) (Fig. 1b). Although most of the other factors individually did not alter the positive biodiversity effect on invasion resistance, it was strengthened when multiple factors were imposed simultaneously (two co-acting factors: mean ΔNBE = 0.16, 95% CI = [0.05, 0.27]; three co-acting factors: mean ΔNBE = 0.64, 95% CI = [0.37, 0.92]) (Fig. 1b).

The strengthened biodiversity effect on invasion resistance under warming conditions was also indicated by the finding that warming had a negative effect on invasion resistance of monocultures but not on invasion resistance of mixtures (Supplementary Figs. 2, 3a). Similarly, the weakened biodiversity effect on invasion resistance under drought conditions was consistent with the finding of a positive effect of drought on invasion resistance of monocultures and the absence of such an effect in mixtures (Supplementary Figs. 2, 3b). While eutrophication did not alter the biodiversity effect on invasion resistance, it decreased the invasion resistance of both monocultures and mixtures (Supplementary Figs. 2, 3c). In addition, while three co-acting factors increased invasion resistance, the effect was stronger for mixtures than monocultures (Supplementary Figs. 2, 3d), which was consistent with the strengthened biodiversity effect on invasion resistance when there were three co-acting factors.

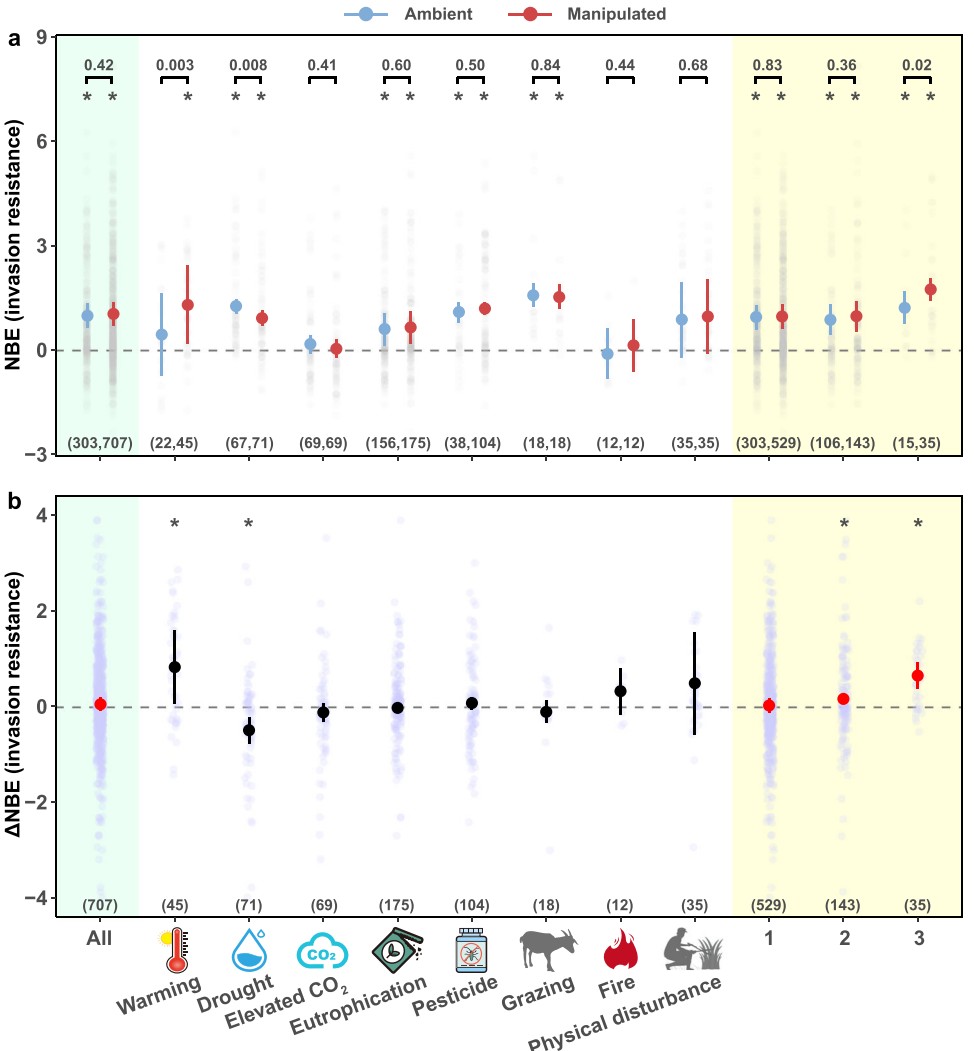

**Fig. 1 | Biodiversity effects on invasion resistance under multiple environmental changes.** The net biodiversity effect (NBE) on invasion resistance (**a**) and the difference in NBE between ambient and manipulated environmental conditions (ΔNBE) (**b**). Positive values of NBE indicate higher invasion resistance of resident mixtures in comparison with that of resident monocultures, whereas negative values indicate the opposite. Positive values of ΔNBE indicate stronger biodiversity effects under manipulated environmental conditions in comparison with ambient conditions, and vice versa. In panel a, the numbers above the brackets are the *p*-values of the $Q_M$ tests for the effect of environmental manipulation (ambient vs. manipulated) on NBE. The numbers in brackets show the number of effect sizes. Points with error bars are the estimated means with corrected 95% confidence intervals. Confidence intervals not overlapping with the dashed line (i.e. 0) indicate statistical significance, as indicated by asterisks. Green shading indicates the analysis on all environmental change factors and yellow shading indicates the analysis on different numbers of factors. Symbols of environmental change factors are created by Yue Chen.

For the subset of studies in which we could distinguish between internal invaders and external invaders (either native or alien), we found that the positive effect of biodiversity on invasion resistance was strongest for external aliens (Supplementary Fig. 4; Supplementary Table 1). However, we found no significant interaction between environmental manipulation and invader type (Supplementary Table 1), suggesting that the biodiversity effect on the resistance to internal and external invaders was consistent across environmental conditions.

We found that the biodiversity effect on invasion resistance was positively associated with the effect on resident productivity across various environmental conditions (Fig. 2). Moreover, the biodiversity effect on invasion resistance increased with the number of resident species in the mixture (Fig. 3), but had overall weak relationships with experimental duration and experimental unit size when the different factors were analyzed (Supplementary Figs. 5–7).

Several environmental change factors had significant net effects on the productivity of resident monocultures (Fig. 4a). Specifically, monoculture productivity was increased on average by warming and eutrophication, indicating that these were favorable environmental conditions, but decreased by grazing and three co-acting factors, indicating that these were stressful environmental conditions. Across environmental change factors, invasion-resistance ΔNBE increased when factors caused stressful environmental conditions ($Q_M = 6.29$, *p* = 0.01; Fig. 4b).

## Discussion

While the positive relationship between diversity and invasion resistance was proposed more than 60 years ago[9], and has been well corroborated by experimental studies in grassland systems[20–22], evidence for neutral and even negative relationships has also been reported[25,26]. Our meta-analysis of 25 factorial grassland experiments showed that

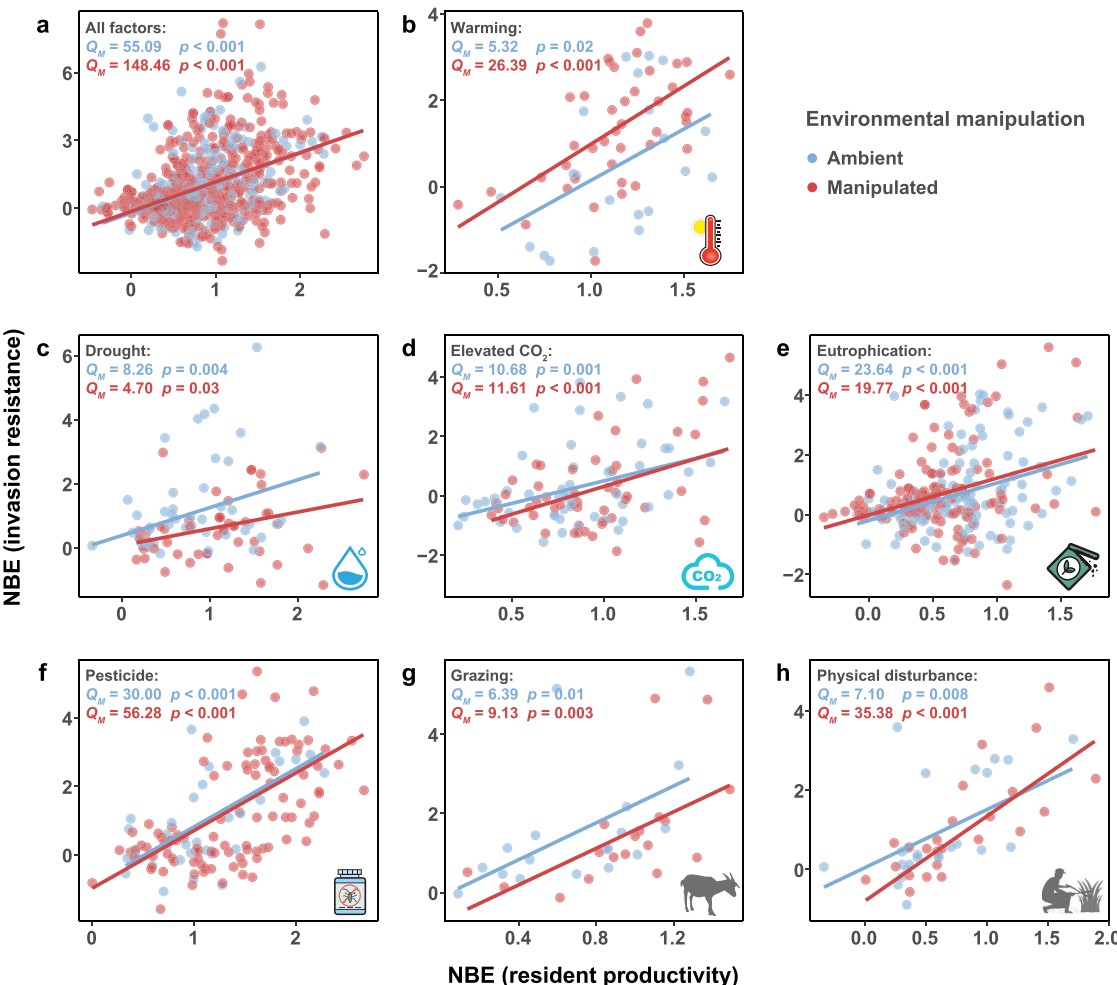

**Fig. 2 | Relationships between the net biodiversity effect (NBE) on invasion resistance and the effect on resident productivity.** Relationships were tested using the $Q_M$ tests for datasets of all environmental change factors (**a**), warming (**b**), drought (**c**), elevated $CO_2$ (**d**), eutrophication (**e**), pesticide (**f**), grazing (**g**) and physical disturbance (**h**). Positive values of NBE indicate higher invasion resistance or productivity of resident mixtures in comparison to resident monocultures, whereas negative values indicate the opposite. Blue indicates the ambient condition and red indicates the manipulated environmental condition. Symbols of environmental change factors are created by Yue Chen.

these seemingly conflicting patterns can, at least partially, be explained by the dependence of the relationship between diversity and invasion resistance on environmental conditions. Specifically, while our results generally supported the hypothesis that plant diversity promotes invasion resistance of grassland communities, we found that the type and number of environmental change factors could modulate the strength of the positive biodiversity effect on invasion resistance.

We found that across all environmental change factors, invasion-resistance ΔNBE increased when factors caused stressful environments, which is in line with the stress-gradient hypothesis that predicts stronger biodiversity effects in more stressful environments[28,31]. However, there were exceptions for particular factors. For example, although warming resulted in a favorable environment, it strengthened the biodiversity effect on invasion resistance. As there was no significant influence of warming on the biodiversity effect on resident productivity (Supplementary Fig. 8), this result may be because alien plant species benefit more from elevated temperatures than native plant species[32], thereby decreasing invasion resistance of monocultures under warming conditions (Supplementary Fig. 2a). A recent study also showed that plant diversity buffered elevated temperature in grasslands[41], which could reduce the positive impact of warming on invaders and resulted in a strengthened biodiversity effect on invasion resistance under warming conditions (Supplementary Fig. 3a).

Drought, on the other hand, resulted in a stressful environment, but nevertheless weakened the biodiversity effect on invasion resistance. However, our finding that drought strengthened the biodiversity effect on resident productivity is consistent with the prediction of the stress-gradient hypothesis (Supplementary Fig. 8). This discrepancy may be because alien plant species suffered more from drought than native plant species[32], thereby increasing invasion resistance of monocultures under drought conditions (Supplementary Fig. 2a). In contrast, diverse plant communities have denser canopies that reduce solar radiation at the soil level and thereby reduce evaporation[42,43]. This could buffer the negative impact of drought on invaders and result in a weakened biodiversity effect on invasion resistance under drought conditions (Supplementary Fig. 3b). Taken together, our results suggest that warming and drought altered the biodiversity effect on invasion resistance through changes in invasion resistance of monocultures, which aligns with previous studies reporting that biodiversity contributes to the stability of ecosystem functions in grassland systems[44,45].

Most studies on the consequences of environmental change factors for biodiversity effects have focused on single factors[39,46]. Here, however, we were able to examine the joint effects of co-acting factors and found that the positive biodiversity effect on invasion resistance became stronger as the number of factors increased. This result is

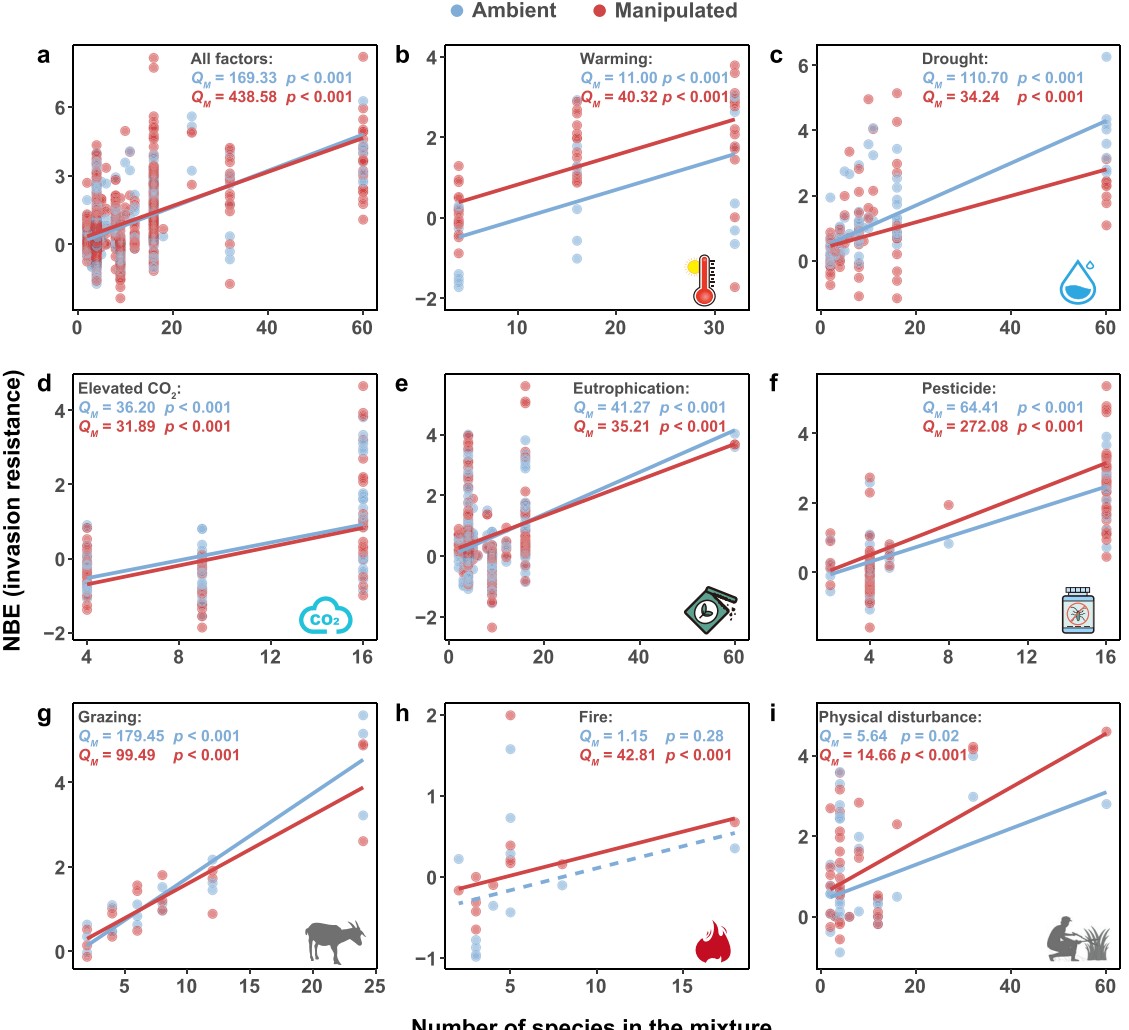

**Fig. 3 | Relationships between the net biodiversity effect (NBE) on invasion resistance and number of species in the mixture.** Relationships were tested using the $Q_M$ tests for datasets of all environmental change factors (**a**), warming (**b**), drought (**c**), elevated $CO_2$ (**d**), eutrophication (**e**), pesticide (**f**), grazing (**g**), fire (**h**) and physical disturbance (**i**). Positive values of NBE indicate higher invasion resistance of resident mixtures in comparison to resident monocultures, whereas negative values indicate the opposite. Blue indicates the ambient condition and red indicates the manipulated environmental condition. Symbols of environmental change factors are created by Yue Chen.

consistent with a recent meta-analysis reporting that plant communities were more likely to be altered when facing at least three global change factors simultaneously[47]. Our finding that three co-acting factors strengthened the biodiversity effect on invasion resistance is also consistent with the prediction of the stress-gradient hypothesis. While each of the individual factors resulted in either stressful, favorable or unaltered environments, three co-acting factors caused a stressful environment (Fig. 4a). This aligns with a recent study reporting that synergistic interactions between co-acting factors significantly decreased the performance of a herbaceous plant (i.e. resulted in a stressful environment)[38]. Although three co-acting factors increased the invasion resistance of both monocultures and mixtures, they had larger impacts on the invasion resistance of mixtures than of monocultures (Supplementary Figs. 2, 3d). This suggests that environmental change factors acted synergistically and increased complementarity effects in mixtures[48,49], which increased resistance against invasion.

Our finding that biodiversity effects on invasion resistance and resident productivity were positively associated suggests that plant diversity effects on resident productivity −and the associated greater competitive ability− may be a mechanism by which resident communities resist invasion. This is also supported by the results from a number of individual studies in grassland communities[20,24,39]. However, as it has been found that biodiversity effects frequently increase over time, primarily through an increase of complementary effects[50,51], we surprisingly found only weak relationships between the biodiversity effect on invasion resistance and experimental duration. Given the larger maximum experimental duration in our meta-analysis (~24 years) compared to other grassland (~15 years)[50] and forest (~8 years)[51] studies, this discrepancy is likely explained by the negligible role of complementarity effects in our study. This is indicated by the fact that there was little transgressive resistance (an indicator of complementarity effects) of biodiversity to invasion (Supplementary Fig. 9), suggesting that the observed biodiversity effect was primarily due to selection effects. This result aligns with previous meta-analyses demonstrating that in most experiments, the most diverse communities did not achieve greater biomass than the single most productive species[52,53].

Our findings may have implications for grassland management aimed at reducing plant invasions under continued global environmental change. First, our result that mixtures, in contrast to monocultures, did not experience a negative effect of warming on invasion resistance (i.e. resulting in a stronger biodiversity effect on invasion

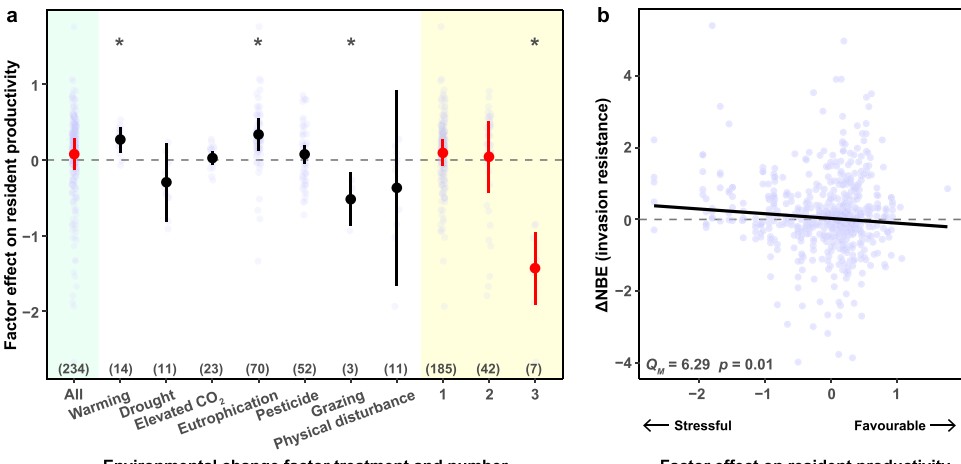

**Fig. 4 | Effects of environmental change factors on the productivity of resident monocultures and their relationship with the difference in the net biodiversity effect on invasion resistance between ambient and manipulated environmental conditions ($\Delta$NBE).** In panel **a**, positive values of the factor effect on resident productivity indicate that environmental change factors increase the productivity of resident monocultures and thus provide a favorable condition, and negative values indicate that environmental change factors decrease the productivity of resident monocultures and thus provide a stressful condition. The numbers in brackets show the number of effect sizes. Points with error bars are the estimated means with corrected 95% confidence intervals. Confidence intervals not overlapping with the dashed line (i.e. 0) indicate statistical significance, as indicated by asterisks. Green shading indicates the analysis on all environmental change factors and yellow shading indicates the analysis on different numbers of factors. In panel **b**, the relationship between $\Delta$NBE and the factor effect on resident productivity was tested using the $Q_M$ test.

resistance), suggests that biodiversity has a buffering effect. This implies that maintaining and enhancing native plant diversity should be a priority to prevent invasion by alien species in an increasingly warmer world. Although drought had positive effects on invasion resistance of monocultures, this was not the case for mixtures. Nevertheless, the relationship between biodiversity and invasion resistance was still positive under drought, indicating that biodiversity is also important under drought. While eutrophication did not alter the biodiversity effect on invasion resistance, its negative impacts on invasion resistance, irrespective of the diversity of resident communities, suggest that grassland managers should reduce the use of fertilizer that may promote plant invasions. Furthermore, our result that plant diversity strengthened the positive effect of three co-acting factors on invasion resistance (i.e. had a stronger biodiversity effect on invasion resistance), suggests that enhancing plant diversity should be prioritized to increase resistance of grasslands against invasion in a changing world in which plant communities may be exposed to multiple factors simultaneously.

Our meta-analysis has several caveats. First, like many meta-analyses, we found evidence for publication bias in our dataset (Supplementary Fig. 10), likely because studies with low precision that found a negative relationship between diversity and invasion resistance —which contradicts the expected positive relationship— are difficult to publish. Nevertheless, because our study was primarily focused on how environmental change factors modulate the strength of biodiversity effects, this bias should not influence the main conclusions drawn from our study. Indeed, our sensitivity analysis indicated that the publication bias was unlikely to influence the robustness of our conclusions (Supplementary Fig. 11). Second, our main finding that biodiversity consistently increased invasion resistance under environmental change factors is based on the performance of all invaders. Despite this, our results could also have implications for biodiversity conservation in an increasingly invaded world, because our subset analysis showed that alien invaders were the most strongly resisted by biodiversity, and that the biodiversity effect on the resistance to different types of invaders was consistent across environmental conditions. Third, while our search aimed to include all taxa and ecosystem types, we mainly found suitable data on the relationship between plant diversity and invasion

resistance in grassland systems. Whether our findings are applicable to other ecosystems (e.g. forests) and other taxa (e.g. microbes and phytoplankton) remains unclear and should be explored further in future studies. Finally, the number of experiments included in our meta-analysis was relatively small, which was especially evident for particular factors (i.e. elevated $CO_2$ and grazing). We acknowledge that this could result from the complex and large experimental designs that are required to simultaneously manipulate biodiversity, invasion and environmental change factors. Nevertheless, the studies that made these three types of manipulations suggest that more attention should be paid to the relationship between diversity and invasion resistance in a rapidly changing world. Furthermore, amongst the studies we analyzed, the number of simultaneously applied factors and their combination was limited. This calls for experiments that incorporate more combinations of co-acting factors to explore potential generality and/ or variation in higher-order interactions of factors on the relationship between diversity and invasion resistance.

## Methods
### Data compilation
We compiled a dataset that included factorial experiments that manipulated species richness together with at least one of several environmental change factors. We followed the PRISMA protocol[54] to identify, select and synthesize studies (Supplementary Fig. 12). Specifically, we searched the ISI Web of Science database, with no restriction on publication year, using the following search terms: (species richness OR diversity OR biodiversity) AND (invasion resistance OR biotic resistance OR invasibility) AND (global change* OR climate change* OR anthropogenic stressor* OR warm* OR temperatur* OR heat* OR drought OR water* OR precipitation OR rain* OR carbon dioxide OR $CO_2$ OR nutrient* OR fertiliz* OR fertilis* OR eutroph* OR pollution OR biocid* OR pesticid* OR fungicid* OR insecticide* OR herbicid* OR bacteriacid* OR nematicide* OR graz* OR herbivor* OR trampl* OR disturb* OR mow* OR clip* OR burn* OR fire*) AND (manipulat* OR treat* OR experiment*). We also searched for additional studies that were included in previous meta-analyses on the relationship between diversity and invasion resistance[15,17,39], as well as the online repositories of two large biodiversity experiments: the Jena experiment in Germany

(https://jexis.idiv.de/) and the Cedar Creek experiment in the United States (https://www.cedarcreek.umn.edu/research/data).

We conducted the initial search on 10 August 2023, yielding a sample of 2096 publications. Of these, 43 duplicates were discarded, resulting in 2053 publications after the first phase of screening. After the removal of publications that based on the titles and abstracts were review or modeling studies, we assessed the remaining 1652 papers for eligibility of inclusion in our analysis using the following criteria: (1) the study must have manipulated the number of species in the resident community directly (i.e. observational studies were excluded); (2) the study must have compared mixtures with monocultures under both ambient and manipulated environmental conditions; (3) the study must provide the mean, statistical variation (standard deviation, standard error or 95% confidence intervals), and sample sizes for the performance of invaders (including both alien and native species) in different treatments. Together with five studies obtained from the online repositories of the Jena and Cedar Creek experiments, we found a total of 25 studies that met these criteria (Supplementary Data 1). All of these studies focused on herbaceous plant communities grown under natural or semi-natural conditions, except for two that were conducted in the greenhouse (excluding these two greenhouse studies did not qualitatively affect our conclusions; Supplementary Fig. 13). Environmental change factors included warming ($N = 2$), drought ($N = 5$), elevated atmospheric $CO_2$ ($N = 1$), eutrophication ($N = 14$), pesticide use (e.g. fungicide and insecticide) ($N = 4$), grazing by domestic animals ($N = 1$), human-caused fire ($N = 2$), physical disturbance (e.g. mowing and trampling) ($N = 5$), and combinations of two ($N = 7$) or three ($N = 1$) of these factors.

We used the performance −measured as biomass or percent cover− of all invaders as a proxy of invasion resistance of the resident community. Specifically, a lower performance of invaders indicates a higher invasion resistance of the resident community. If information about the identity of the invader was provided, we also distinguished between internal invaders of an experimental unit that were residents of other units of the experiment and external invaders that were not part of the experiment's resident species pool. For the latter, we also distinguished between native and alien invaders (non-native to the location where the experiment was done). When several performance metrics were reported in the same study (e.g. cover and biomass), we used only the biomass of invaders because the majority of the studies (17 of 25) only reported biomass data. We also found that excluding the four studies that only reported cover data did not qualitatively affect our conclusions (Supplementary Figs. 14–16). We extracted the mean, statistical variation, and sample size for the performance metrics of invaders directly from data appendices, the text or tables, or from the figures using GetData Graph Digitizer (version 2.20, Russian Federation). When the relevant data were not provided in the publication, we contacted the corresponding author to obtain them. In total, we compiled a dataset consisting of 1010 observations on the performance of invaders at different levels of resident diversity. In addition to the performance of invaders, we also extracted data on the productivity (biomass or cover) of the resident community, resident species richness (1–60), experimental unit size (0.01–47.5 m$^2$) and experimental duration (0.25–24 years) wherever possible.

## Effect size calculation
We calculated the effect size of NBE on invasion resistance, at each diversity level of the resident community under both ambient and manipulated environmental conditions, using the natural log of the response ratio[55]:

$$NBE = \ln(X_{mono}/X_{mix}), \qquad (1)$$

where $X_{mono}$ and $X_{mix}$ are the mean performance of invaders grown in resident monocultures and mixtures, respectively. Positive values of

NBE indicate a higher invasion resistance of resident mixtures than in resident monocultures, whereas negative values indicate the opposite. The variance of NBE, $v_{NBE}$, was calculated as[55]:

$$v_{NBE} = \frac{(S_{mono})^2}{n_{mono} \times (X_{mono})^2} + \frac{(S_{mix})^2}{n_{mix} \times (X_{mix})^2}, \qquad (2)$$

where $S$ is the standard deviation and $n$ is the sample size; and the subscripts 'mono' and 'mix' refer to resident monocultures and mixtures, respectively. For 16 studies with data on resident productivity, we also calculated resident-productivity NBE using Eq. (1), but replaced invaders with the resident community.

To quantify the response of the biodiversity effect to environmental change factors, we calculated the difference in invasion-resistance NBE between ambient and manipulated environmental conditions ($\Delta$NBE), pairwise for each diversity level of the resident community, using the following equation[46]:

$$\Delta NBE = NBE_M - NBE_A, \qquad (3)$$

where the subscripts 'A' and 'M' refer to ambient and manipulated environmental conditions, respectively. Positive values of $\Delta$NBE indicate stronger biodiversity effects under manipulated environmental conditions than under ambient conditions, while negative values indicate the opposite. The variance of $\Delta$NBE, $v_{\Delta NBE}$, was calculated as[46]:

$$v_{\Delta NBE} = (v_{NBE})_A + (v_{NBE})_M. \qquad (4)$$

We quantified the effect of environmental change factors on invasion resistance and its variance in resident monocultures and mixtures, respectively, using the following equations[55]:

$$\text{Factor effect on invasion resistance} = \ln(X_A/X_M), \qquad (5)$$

$$v_{\text{Factor effect}} = \frac{(S_A)^2}{n_A \times (X_A)^2} + \frac{(S_M)^2}{n_M \times (X_M)^2}, \qquad (6)$$

where $X_A$ and $X_M$ are the mean performance of invaders under ambient and manipulated environmental conditions, respectively. Positive values of the factor effect on invasion resistance indicate that environmental change factors increase invasion resistance, whereas negative values indicate the opposite.

To explore whether environmental change factors result in stressful or favorable environments, we quantified the effect of environmental change factors on the productivity of resident monocultures using the following equation[55]:

$$\text{Factor effect on resident productivity} = \ln(Y_{mono,M}/Y_{mono,A}), \qquad (7)$$

where $Y_{mono,A}$ and $Y_{mono,M}$ are the mean productivity of resident monocultures under ambient and manipulated environmental conditions, respectively. Positive values of the factor effect on resident productivity indicate that the environmental change factor increases the productivity of resident monocultures and thus is a favorable condition for plant growth, while negative values indicate that the environmental change factor decreases the productivity of resident monocultures and thus provides a stressful condition[46]. We only considered data for monocultures because biodiversity might buffer the effect of environmental change factors in the mixtures[46].

## Statistical analyses
Because biodiversity effects are scale-dependent and sensitive to species richness and duration of the experiment[56,57], we used meta-regression models that included resident species richness,

experimental duration and experimental unit size as covariates to test the effect of environmental manipulation (ambient vs. manipulated) on NBE and to derive the mean effect size of ΔNBE. We first performed these analyses for all environmental change factors and then for different types or numbers of factors. For the subset of studies with data on the invader type (internal invader, native external invader, alien external invader), we included the interaction between environmental manipulation and invader type in meta-regression models to explore whether invader type influences the effect of environmental change factors. Since we calculated the effect size of NBE by comparing multiple diversity levels to the same monoculture control, we accounted for this non-independence by computing the variance-covariance matrix of effect sizes[58]. The inverse of the sampling variance of the variance-covariance matrix was then used to weight the precision of effect sizes. To further account for possible non-independence of observations from the same study and for between-observation errors, we included observations nested in "study" as random factors in models[59].

To test whether the biodiversity effect on invasion resistance is associated with the effect on resident productivity, resident species richness, experimental duration and experimental unit size, we used meta-regression models that included these moderators under ambient and manipulated environmental conditions, respectively. We used meta-regression models that included experimental duration and experimental unit size as covariates to test the effect of environmental change factors on invasion resistance or productivity of resident monocultures. To test the effect of environmental change factors on invasion resistance of resident mixtures, we also included resident species richness as covariate in meta-regression models. Furthermore, we used meta-regression models that included species richness, experimental duration and experimental unit size as covariates to test the relationship between invasion-resistance ΔNBE and the factor effect on resident productivity.

Finally, we tested publication bias in two ways[60]: (1) visual inspection for asymmetry in the funnel plot of the residuals from the meta-regression models, and (2) testing funnel asymmetry using Egger's regression by including sampling standard error as a moderator in the meta-regression models (a significant sampling standard error indicates asymmetry in the funnel). When publication bias was detected, we conduced sensitivity analysis to identify potential outliers based on the Cook's distance[61] and then conducted the analyses after removing outliers.

We performed all statistical analyses in R 4.1.3[62]. Meta-regression analyses were performed using the 'rma.mv' function in the 'metafor' package (version 4.1–0)[63]. We conducted the $Q_M$ test to determine the significance ($p < 0.05$) of moderators using the 'anova' function in the 'metafor' package. We estimated the mean effect size of the biodiversity effect or the factor effect from the meta-regression models and corrected the 95% CI using the Bonferroni method with the 'emmeans' package (version 1.8.4–1)[64]. We considered the mean effect size to be significant if the corrected 95% CI did not overlap zero. We tested pairwise differences in the mean effect size of the biodiversity effect among invader types using the 'multcomp' package (version 1.4–22)[65].

### Reporting summary
Further information on research design is available in the Nature Portfolio Reporting Summary linked to this article.

## Data availability
All raw data are archived in Figshare at https://doi.org/10.6084/m9. figshare.24953433[66]. Source data are provided with this paper.

## Code availability
All codes are archived in Figshare at https://doi.org/10.6084/m9. figshare.24953433[66].

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

## Acknowledgements

The study was funded by National Key Research and Development Program of China (2022YFC2601100), National Natural Science Foundation of China (32030067), Department of Science and Technology of Yunnan Province (202405AS350011) and Talent Scientific Fund of Lanzhou University awarded to J.W. and B.L. We thank Yue Chen for drawing the symbols of environmental change factors. We also thank the authors who generously shared their data. The data on the Jena Experiment were obtained by C.C. from the Jena Experiment database (https://jexis.idiv.de/) in August 2023. We thank Anja Vogel, Alexandra Weigelt and Anne Ebeling for making this data set available. The Jena Experiment is a research unit funded by the Deutsche Forschungsgemeinschaft (FOR 456/1451/5000).

## Author contributions

J.W. conceived the study. C.C. led the data collection, with help from Z.L., W.S., and X.C. C.C. analyzed the data and wrote the manuscript, with substantial input from M.v.K., J.W., B.L., and Z.Z.

## Competing interests

The authors declare no competing interests.
