## [Peer Review File · Nature Communications]

Biodiversity increases resistance of grasslands against plant invasions under multiple environmental changesREVIEWER COMMENTS

Reviewer #1 (Remarks to the Author):

The study by Cheng and colleagues is considering an important and timely topic, namely is resistance against invasions modulated by biodiversity and altered by global change. The authors are testing this question using a meta-analytical framework to compare how the biodiversity effect is changed by different global change drivers and test if the number of global change drivers also plays a role in affecting the resistance of plant communities to invasions. This paper is considering experimental studies. Therefore, the causality between driver and response is clear, which is a big bonus compared to previous meta-analysis on this topic that have been using correlative data where causality is not always clear. This study therefore will significantly improve our knowledge in this field. Overall, I think this is a well conducted and sound meta-analysis. The sample size is not very high (25 studies) but I think there are just not many experiments that are testing this complex but important research question (i.e. experiments that simultaneously manipulate biodiversity, invasions and global change). The manuscript is very well written and the introduction and discussion are clear.

I only have some minor issues/questions that I would like to see the authors address in a revised version of this paper:

Methods

- Why did the authors not include "BEF experiment" and/or "biodiversity-ecosystem functioning experiment" as search term in their literature review? As far as I understood these are the kind of experiments that the authors are looking for but I could not find this in their search terms.
- How much did the plot size vary across the experiments and did it affect the response? Did the authors test if i) there is covariation between the plot size and the effect size, replication and variance, and ii) covariation between plot size, replication and variance and the moderators of your meta-analysis? See paper by Spake et al. 2021 [attached]
- L 259 It would be good to also mention how many replicates there are per ECF

Discussion

L 147: as the stress gradient hypothesis is prominently discussed here it would be useful if it could be mentioned in the introduction. This would also help the reader follow the whole section on stress intensity (see comment below).

L 151/158 negative stress intensity means less stress? It is a little confusing to me to use positive and negative stress intensity, as stress is considered to be a negative effect of e.g. an environmental factor on the performance (as the authors define it in the methods part). I think a different wording, e.g. benign/favorable vs stressful or something similar, would make it easier for the reader to understand what the authors mean.

L 150-164 It is hard to follow this whole section on stress intensity. Why is it surprising that if warming increases plant growth (in monocultures) i.e. it makes the environment more benign for resident plants, that it also strengthens the biodiversity effect and less species are able to invade? Similarly, why is it surprising that drought which leads to more stressful conditions weakens the biodiversity effect in invasion resistance? Could the relative importance of environmental filtering vs. competition play a role here?

L190 Transgressive resistance was never mentioned or introduced in the main text (but I found it in the supplement). I would find it useful if this term would be introduced earlier or better explained in this section.

L 206 vegetation/community type/ecosystem type instead of "biomes"? As far as I know biome considers the biogeographical extend of the potential vegetation due to certain climatic conditions rather than the realized vegetation. IN the next sentence the authors also refer to ecosystems.

Figures

Fig. 3: Please consider changing the axis title. What is meant by number of species in excess of the monocultures? Species richness in mixtures? Species richness in mixtures minus species richness in monocultures?

Reviewer #2 (Remarks to the Author):

This paper reconfirmed the biotic resistance hypothesis (native diversity reduces invasion) across grasslands. The interesting angle is to test this under different global change drivers, and what the synergistic effects are across multiple global change drivers. However, it really lacks sound theory/hypotheses around what might be expected and why (with regard to specific stressors, multiple stressors, why range expanders v novel invaders may have different patterns). It is definitely possible I missed something, but this should be made much stronger/compelling. This makes it difficult to judge whether the work is of significance and what inferences can be made. The data analyses seem reasonable to me. There is enough detail provided for reproducibility, but I did not get access to the data itself. There were several points in the results where I was not expecting specific results because it wasn't clear this was linked to the three questions in the introduction. Overall, I think the concept of testing biotic resistance across human stressors is interesting, but I think the paper needs stronger and clearer hypotheses, particularly based on other grassland studies.

[See detailed comments in the attachment]

Developing hypotheses and background support: The general hypotheses/argument for why environmental stress will weaken biotic resistance is weak. Simply because a process favors invasives/non-natives over native species does not entail that the relationship will weaken. It would just shift the window over which one is analyzing the trend. As a broader point, I would like to see more congruency between outlined hypotheses and the results. For example, there are three questions in the introduction, but none involve or define stress, and this comes somewhat as a surprise later. Even with regard to other questions, it is not so clear what the expected outcome/hypothesis is.

Scope transparency: Please be clear that this work is only across grassland ecosystems. Further, I would suggest you focus in on developing hypotheses and background from research in this system. In the discussion, you could then compare and contrast the study results to previous work across grasslands. In addition, you may contrast your results with results found in other biomes. Ultimately, the grassland nature of this meta-analysis should be much more explicit in your hypotheses, discussions and interpretations.

Title: The title is misleading and should specify in grasslands.

Abstract generally: please somehow clarify what type on invasion you refer to (invasion occurrence, severity in abundance, basal area, harm).

L 22. Change “species richness” to “plant species richness”

L 26-27. I think it would be helpful to specify when/where the relationship is positive (not only strengthening or weakening) in the abstract/overall conclusions. Are relationships always positive? *No, not with drought.

L 27-28. This conclusion is kind of unsatisfying. Is there a way to generalize why certain types result in an increase and certain in a decrease? This would really aid in understanding what direction certain ecosystems would go toward.

L 29-30. I suggest a concluding statement that is more directly related to how the information gained related to changes in the biodiversity-invasion relationship with ECFs informs management.

L 39. What does this mean?

L 39 & 40. Magnitude and type?

L 47. Add Beaury et al. (<https://onlinelibrary.wiley.com/doi/full/10.1111/ele.13446>)

L 48-49. They are unable to attribute causation as they are correlative studies. However, this particular study accounts for change in environmental soil and climate variables. I would amend this sentence to clearly note that these are observational studies, which may limit inference of

causality, and have a separate sentence making the point that “both native and alien species respond to variation in the environment and each other”.

L 49-53. I don't really think the “Therefore” here follows so clearly from the previous sentences. Are you saying that global analyses might be biased due to this positive relationship? This is only true I think if you analyze all data as one, but may not be a problem if you have repeated local plot data.

L 53. Here and throughout, you need to define diversity. It can be measured in many ways, but it seems you are saying richness. Also, define invasion resistance. What is the value you are using to measure invasion? Probability of invasion, number of invasive species, proportion of invasive species, basal area of invaders?

L 64. What is biocide pollution?

L 64-70. I don't think this follows. The drought is only reducing native diversity, so will just move species richness down. However, this doesn't mean that the actual line would flatten between native diversity and invaders.

L 70-76. The idea here is that global change favors invaders at the expense of native species. This would by definition reduce native species, and increase invaders, but really does not say anything about the strength of the relationship. If we assume a community has the space for a given number of species, environmental change will simply move the curve/line to the “left” toward lower native and “up” toward higher invasive species. How does this actually predict anything about the strength of the relationship?

L 72. This contrasting result is included, which is good. But, there follows no explanation or hypothesis about why this was found.

L 79. Please add a reference here.

L 94-96. I don't understand what “performance” means. Please define clearly.

L 117-118. There needs to be more justification/theory/hypotheses behind why you separate these two groups. What do we expect?

L 123-124. This also comes as an unanticipated result/analyses. This doesn't align with the three major questions posed at the end of the introduction.

L 124-126. This is cool. What about plot/unit size?

L 128-133. This monoculture analysis kind of comes out of the blue. I could see why it is valuable, but was a bit surprising. It should be integrated better into the introduction/justification.

L 140-142. It needs to be acknowledged that this is comparing everything to grasslands. Actually, I suggest you restructure the background/intro and discussion to compare your results with known grassland results instead of reference your study as testing the entire literature across all systems.

L 146-149. There is not reference to this in the introduction. Please integrate all your hypotheses in the introduction so it is clear what you predict and how your results compare.

L 175-178. This needs to be discussed more thoroughly.

Overall: there needs to be a clearer communication of the analyses/approach before results are reported and discussed. For example, there was no mention of complementarity v. selection analyses.

L 179-181. This sentence is not true. Just because you find a correlation does not point to that driver as the dominant one. Please rephrase.

L 187-189. What is your range of time, especially in comparison to other studies that found it important? If it is relatively short, this may limit your ability to detect a time effect.

L 200-202. I'm not sure exactly why this is a shortcoming. Can you please explain before justifying?

L 214-216. This kind of statement is where you need to acknowledge the appropriate scope of the paper. You looked across grasslands, yet you say "*clear evidence that plant diversity consistently increased invasion resistance of the resident community, mainly through selection effects, across various environmental conditions*". It may be as simple as stating that there is "clear evidence that in grassland systems,..."

L 220-221. What "mechanistic insights"? Maybe instead of this sweeping statement give some concrete examples.

L 256. 25 studies seems exceedingly low. Can you please justify this? What was the main bottleneck in removing studies?

L 263. This is too late to report this within methods, and should be in the main text as well.

L 271. Can you test this both ways? Why would biomass v cover be preferable? This should be justified/explained.

L 277. Did this include again both cover and biomass? Does mixing these impact results?

L 283. This basically tells us the effect size between one and “more” species, but this “more” can differ drastically across studies, meaning that the NBE value can mean very different things in different studies. Can you expand on the range of “more”/”mix” species and potentially explicitly test for the effect of mix# species on the results of NBE? *Looks like you included this later.

L 296. Can you confirm that these values NBE values compared were calculated from the exact same # of mixture species?

L 315- 318. Looks like you included species richness here. However, which species richness you are referring to is unclear. Is this native richness?

Manuscript number: NCOMMS-24-01287

Title: Biodiversity increases resistance against plant invasions under multiple environmental changes

Authors: Cai Cheng, Zekang Liu, Wei Song, Xue Chen, Zhijie Zhang, Bo Li, Mark van Kleunen, Jihua Wu

Responses to reviewers' comments, Blue text = authors' reply, line numbers in blue color correspond to the revised version of the manuscript.

Responses to Reviewer #1

1. The study by Cheng and colleagues is considering an important and timely topic, namely is resistance against invasions modulated by biodiversity and altered by global change. The authors are testing this question using a meta-analytical framework to compare how the biodiversity effect is changed by different global change drivers and test if the number of global change drivers also plays a role in affecting the resistance of plant communities to invasions. This paper is considering experimental studies. Therefore, the causality between driver and response is clear, which is a big bonus compared to previous meta-analysis on this topic that have been using correlative data where causality is not always clear. This study therefore will significantly improve our knowledge in this field. Overall, I think this is a well conducted and sound meta-analysis. The sample size is not very high (25 studies) but I think there are just not many experiments that are testing this complex but important research question (i.e. experiments that simultaneously manipulate biodiversity, invasions and global change). The manuscript is very well written and the introduction and discussion are clear. I only have some minor issues/questions that I would like to see the authors address in a revised version of this paper.

Authors' response (AR): We appreciate the reviewer's positive evaluation and valuable comments on our manuscript. We revised the manuscript according to each of those comments. Please, see our point-by-point responses below.

2. Why did the authors not include "BEF experiment" and/or "biodiversity-ecosystem functioning experiment" as search term in their literature review? As far as I understood these are the kind of experiments that the authors are looking for but I could not find this in their search terms.

AR: We thank the reviewer for this suggestion. Yes, BEF experiments are what we are looking for. Actually, we had initially included "BEF experiment* OR biodiversity-ecosystem functioning experiment*" as search term. When we used "(BEF experiment* OR biodiversity-ecosystem functioning experiment*) AND (invasion resistance OR biotic resistance OR invasibility)" as search terms, 17 of the 29 retrieved publications were already included in our study. We read the remaining 12 publications and found

that none of them met the criteria of our study. Because the inclusion of the “BEF experiment* OR biodiversity-ecosystem functioning experiment*” search term did not provide additional suitable studies, we did not mention the search terms in our manuscript.

3. How much did the plot size vary across the experiments and did it affect the response? Did the authors test if i) there is covariation between the plot size and the effect size, replication and variance, and ii) covariation between plot size, replication and variance and the moderators of your meta-analysis? See paper by Spake et al. 2021 [attached].

AR: We thank the reviewer for these questions and the reference. The experimental unit size ranged from 0.01 (pot) to 47.5 (plot) m², we added this information in the revised manuscript (Lines 352-353). In response to this comment and another one by Reviewer #2, we analyzed the role of experimental unit size and found weak (mostly non-significant) relationships between experimental unit size and the biodiversity effect on invasion resistance (Lines 162-163, Figs. S6, S7).

In the revised manuscript, we also examined the correlations between experimental unit size, experimental duration, resident species richness, replication, effect size and variance of the biodiversity effect. We found that there were weak to moderate correlations between these indices, with correlation coefficients ranging from 0.009 to 0.60 (Fig. S7).

4. L 259 It would be good to also mention how many replicates there are per ECF.

AR: We added the number of replicates per ECF in the revised manuscript (Lines 330-334).

5. L 147: as the stress gradient hypothesis is prominently discussed here it would be useful if it could be mentioned in the introduction. This would also help the reader follow the whole section on stress intensity (see comment below).

AR: We thank the reviewer for this suggestion. We incorporated the stress-gradient hypothesis into the introduction of the revised manuscript (Lines 68-75, 88-91, 125-128).

6. L 151/158 negative stress intensity means less stress? It is a little confusing to me to use positive and negative stress intensity, as stress is considered to be a negative effect of e.g. an environmental factor on the performance (as the authors define it in the methods part). I think a different wording, e.g. benign/favorable vs stressful or something similar, would make it easier for the reader to understand what the authors mean.

AR: Indeed, in the previous version of our manuscript, a negative stress intensity

indicated a less stressful environment. Following your suggestion, we used “favorable vs. stressful” instead of “positive vs. negative” to make our expressions clearer. We also revised the formula by calculating the ECF effect on the productivity of resident monocultures (Lines 387-397). In the revised manuscript, positive values of ECF effect on the productivity of resident monocultures indicate that the ECF results in a more favorable environment, while negative values indicate that the ECF results in a more stressful environment.

7. L 150-164 It is hard to follow this whole section on stress intensity. Why is it surprising that if warming increases plant growth (in monocultures) i.e. it makes the environment more benign for resident plants, that it also strengthens the biodiversity effect and less species are able to invade? Similarly, why is it surprising that drought which leads to more stressful conditions weakens the biodiversity effect in invasion resistance? Could the relative importance of environmental filtering vs. competition play a role here?

AR: The stress-gradient hypothesis predicts stronger biodiversity effects in stressful environments due to weak competition or strong facilitation between species. However, we found that although drought resulted in a stressful environment, it weakened the positive biodiversity effect on invasion resistance. Meanwhile, warming resulted in a favourable environment, yet strengthened the positive biodiversity effect on invasion resistance. Therefore, we are surprised that these results were not consistent with the prediction of the stress-gradient hypothesis.

In order to help explain these results, we further tested the effect of ECFs on invasion resistance of monocultures and mixtures in the revised manuscript (Lines 142-152, 379-386; Figs. S2-S3). We found that invasion resistance of monocultures was reduced by warming but increased by drought. However, these impacts were not present in mixtures. Because we focused on experimental studies that directly controlled plant species richness under ambient and manipulated environmental conditions, we think that there would be little role of environmental filtering.

8. L190 Transgressive resistance was never mentioned or introduced in the main text (but I found it in the supplement). I would find it useful if this term would be introduced earlier or better explained in this section.

AR: We thank the reviewer for this suggestion. We now added an explanation of transgressive resistance in the section indicated by the reviewer (Line 236).

9. L 206 vegetation/community type/ecosystem type instead of “biomes”? As far as I know biome considers the biogeographical extend of the potential vegetation due to certain climatic conditions rather than the realized vegetation. IN the next sentence the authors also refer to ecosystems.

AR: Following this comment, we replaced “biomes” with “ecosystem types” (Line 270).

10. Fig. 3: Please consider changing the axis title. What is meant by number of species in excess of the monocultures? Species richness in mixtures? Species richness in mixtures minus species richness in monocultures?

AR: In the previous version of our manuscript, number of species in excess of the monocultures represents species richness in mixtures minus one (i.e. species richness in monocultures). In order to avoid confusion, we replaced it with number of species in mixtures and redid the analyses (Fig. 3). This did not change our results.

Responses to Reviewer #2

1. This paper reconfirmed the biotic resistance hypothesis (native diversity reduces invasion) across grasslands. The interesting angle is to test this under different global change drivers, and what the synergistic effects are across multiple global change drivers. However, it really lacks sound theory/hypotheses around what might be expected and why (with regard to specific stressors, multiple stressors, why range expanders v novel invaders may have different patterns). It is definitely possible I missed something, but this should be made much stronger/compelling. This makes it difficult to judge whether the work is of significance and what inferences can be made. The data analyses seem reasonable to me. There is enough detail provided for reproducibility, but I did not get access to the data itself. There were several points in the results where I was not expecting specific results because it wasn't clear this was linked to the three questions in the introduction. Overall, I think the concept of testing biotic resistance across human stressors is interesting, but I think the paper needs stronger and clearer hypotheses, particularly based on other grassland studies.

AR: We appreciated the reviewer's encouragements and constructive comments on our manuscript. We revised the manuscript according to each of those comments. In the revised manuscript, we strengthened the theoretical basis and used this and findings from grassland studies to develop four clear hypotheses (Lines 68-77, 88-91, 95-110, 120-128). Please, see our point-by-point responses below.

2. Developing hypotheses and background support: The general hypotheses/argument for why environmental stress will weaken biotic resistance is weak. Simply because a process favors invasives/non-natives over native species does not entail that the relationship will weaken. It would just shift the window over which one is analyzing the trend. As a broader point, I would like to see more congruency between outlined hypotheses and the results. For example, there are three questions in the introduction, but none involve or define stress, and this comes somewhat as a surprise later. Even with regard to other questions, it is not so clear what the

expected outcome/hypothesis is.

AR: We thank the reviewer for pointing this out. In the revised manuscript, we developed hypotheses on how the biodiversity effect on invasion resistance may change in response to environmental change factors (ECFs) based on the stress-gradient hypothesis (Lines 68-75, 88-91, 125-128). We also strengthened the background support for the hypotheses on the relationships between the biodiversity effect on invasion resistance and other moderators (e.g. the biodiversity effect on resident productivity, resident species richness, experimental duration, experimental unit size) and on the role of invader types (Lines 95-110).

3. Scope transparency: Please be clear that this work is only across grassland ecosystems. Further, I would suggest you focus in on developing hypotheses and background from research in this system. In the discussion, you could then compare and contrast the study results to previous work across grasslands. In addition, you may contrast your results with results found in other biomes. Ultimately, the grassland nature of this meta-analysis should be much more explicit in your hypotheses, discussions and interpretations.

AR: Thanks for the great suggestion. In the revised manuscript, we explicitly specified the grassland scope of our study in the title (Line 1) and abstract (Lines 20, 26-27). In the introduction, we developed hypotheses based on grassland studies (e.g. Lines 88-91, 95-110), and in the discussion we compared our results with other grassland studies and with forest studies (e.g. Lines 191-194, 200-202, 204-207, 227-228, 232-234) and we discussed the implications for grassland management (Lines 241-256).

4. Title: The title is misleading and should specify in grasslands.

AR: We added “grasslands” to the title (Line 1).

5. Abstract generally: please somehow clarify what type on invasion you refer to (invasion occurrence, severity in abundance, basal area, harm).

AR: In the abstract of the revised manuscript, we defined invasion as the measurements of biomass or cover of invaders (Lines 21-22).

6. L 22. Change “species richness” to “plant species richness”

AR: Revised as suggested.

7. L 26-27. I think it would be helpful to specify when/where the relationship is positive (not only strengthening or weakening) in the abstract/overall conclusions. Are relationships always positive? *No, not with drought.

AR: We thank the reviewer for this suggestion. We found that the relationship between diversity and invasion resistance was positive averaged across all experiments. While drought weakened the positive biodiversity effect on invasion resistance, we still found a significant positive relationship between diversity and invasion resistance under drought conditions. Based on this comment and comments #8 and #9, we revised the concluding sentence to “Overall, we show that biodiversity helps grassland communities resist plant invasions under multiple environmental changes. Therefore, investment in the protection and restoration of native biodiversity is not only important for prevention of invasions under current conditions but also under continued global environmental change.” (Lines 26-30).

8. L 27-28. This conclusion is kind of unsatisfying. Is there a way to generalize why certain types result in an increase and certain in a decrease? This would really aid in understanding what direction certain ecosystems would go toward.

AR: As mentioned in our response to comment #7, we rephrased the concluding sentence (Lines 26-30). Although we did not generalize why certain types resulted in an increase and others in a decrease in the abstract due to the abstract word limit, we discussed these results and potential implications for grassland management in the discussion (Lines 183-207, 213-223, 241-256).

9. L 29-30. I suggest a concluding statement that is more directly related to how the information gained related to changes in the biodiversity-invasion relationship with ECFs informs management.

AR: As mentioned in our responses to comments #7 and #8, we rephrased the concluding sentence to briefly summarize the management implications of the effects of ECFs on the diversity–invasion resistance relationship (Lines 26-30). In addition, we discussed in detail the implications of our results for grassland management in the discussion (Lines 241-256).

10. L 39. What does this mean?

AR: This means “background climatic conditions”. In the revised manuscript, we replaced “background environmental conditions” with “background climatic conditions” to avoid confusion (Line 39).

11. L 39 & 40. Magnitude and type?

AR: Yes, this expression is more accurate. Revised as suggested.

12. L 47. Add Beaury et al. (<https://onlinelibrary.wiley.com/doi/full/10.1111/ele.13446>)

AR: Revised as suggested.

13. L 48-49. They are unable to attribute causation as they are correlative studies. However, this particular study accounts for change in environmental soil and climate variables. I would amend this sentence to clearly note that these are observational studies, which may limit inference of causality, and have a separate sentence making the point that “both native and alien species respond to variation in the environment and each other”.

AR: We thank the reviewer for this suggestion. We revised this sentence accordingly to clearly state that the observational studies allow limited inference of causality, and we started a separate sentence as suggested (Lines 46-50).

14. L 49-53. I don't really think the “Therefore” here follows so clearly from the previous sentences. Are you saying that global analyses might be biased due to this positive relationship? This is only true I think if you analyze all data as one, but may not be a problem if you have repeated local plot data.

AR: This is a great point, and we replaced “Therefore” with “Given the limited causal inference of observational studies” in the revised manuscript (Line 55).

15. L 53. Here and throughout, you need to define diversity. It can be measured in many ways, but it seems you are saying richness. Also, define invasion resistance. What is the value you are using to measure invasion? Probability of invasion, number of invasive species, proportion of invasive species, basal area of invaders?

AR: In the revised manuscript, we defined diversity as species richness and invasion as biomass or cover of invaders (Lines 57-58).

16. L 64. What is biocide pollution?

AR: It refers to pesticide use. We replaced “biocide pollution” with “pesticide use” throughout the manuscript.

17. L 64-70. I don't think this follows. The drought is only reducing native diversity, so will just move species richness down. However, this doesn't mean that the actual line would flatten between native diversity and invaders.

AR: We thank the reviewer for pointing this out and we realized the shortcoming of this prediction. In the revised manuscript, we rewrote this paragraph of the introduction to incorporate the stress-gradient hypothesis and to predict the effect of ECFs on the relationship between diversity and invasion resistance (Lines 68-75).

18. L 70-76. The idea here is that global change favors invaders at the expense of native species. This would by definition reduce native species, and increase invaders, but

really does not say anything about the strength of the relationship. If we assume a community has the space for a given number of species, environmental change will simply move the curve/line to the “left” toward lower native and “up” toward higher invasive species. How does this actually predict anything about the strength of the relationship?

AR: We revised this part by referring to the stress-gradient hypothesis to support our predictions about potential effect of ECFs on the diversity-invasion resistance relationship (Lines 68-75).

19. L 72. This contrasting result is included, which is good. But, there follows no explanation or hypothesis about why this was found.

AR: In the revised manuscript, we removed this sentence because we restructured this part based on the stress-gradient hypothesis.

20. L 79. Please add a reference here.

AR: Revised as suggested.

21. L 94-96. I don't understand what “performance” means. Please define clearly.

AR: Performance refers to biomass or percent cover. We added this information to this sentence in the revised manuscript (Line 117).

22. L 117-118. There needs to be more justification/theory/hypotheses behind why you separate these two groups. What do we expect?

AR: In the revision, we added the reason why we distinguished between invader types in the introduction and incorporated it into our hypotheses (Lines 105-110, 121-122).

23. L 123-124. This also comes as an unanticipated result/analyses. This doesn't align with the three major questions posed at the end of the introduction.

AR: In the introduction of the revised manuscript, we added the reasons for testing the effect of these moderators and incorporated them into our hypotheses (Lines 95-110, 122-125).

24. L 124-126. This is cool. What about plot/unit size?

AR: In response to this comment and the one by Reviewer #1, we analyzed the role of experimental unit size and found weak correlations (mostly non-significant) between experimental unit size and the biodiversity effect on invasion resistance (Lines 162-163, Figs. S6).

25. L 128-133. This monoculture analysis kind of comes out of the blue. I could see why it is valuable, but was a bit surprising. It should be integrated better into the introduction/justification.

AR: In the introduction of the revised manuscript, we incorporated the stress-gradient hypothesis to justify the monoculture analysis (Lines 68-75).

26. L 140-142. It needs to be acknowledged that this is comparing everything to grasslands. Actually, I suggest you restructure the background/intro and discussion to compare your results with known grassland results instead of reference your study as testing the entire literature across all systems.

AR: In the revised manuscript, we restricted our scope to grasslands in the sentence indicated by the reviewer (Lines 176-179), and we emphasized the grassland scope throughout the manuscript.

27. L 146-149. There is not reference to this in the introduction. Please integrate all your hypotheses in the introduction so it is clear what you predict and how your results compare.

AR: In the revised manuscript, we incorporated the stress-gradient hypothesis into the introduction and our hypotheses (Lines 68-75, 88-91, 125-128).

28. L 175-178. This needs to be discussed more thoroughly.

AR: We rewrote this part to discuss the effect of multiple ECFs more explicitly in the revised manuscript (Lines 213-223).

29. Overall: there needs to be a clearer communication of the analyses/approach before results are reported and discussed. For example, there was no mention of complementarity v. selection analyses.

AR: In the introduction of the revised manuscript, we strengthened the background information for all analyses (e.g. resident plant richness, experimental duration, experimental unit size, complementarity vs. selection effects) (Lines 95-110).

30. L 179-181. This sentence is not true. Just because you find a correlation does not point to that driver as the dominant one. Please rephrase.

AR: Thanks for pointing this out. We rephrased this sentence to tone down this statement (Lines 224-227).

31. L 187-189. What is your range of time, especially in comparison to other studies

that found it important? If it is relatively short, this may limit your ability to detect a time effect.

AR: The experimental duration ranged from 0.25 to 24 years, and we added this information to the methods section of the revised manuscript (Line 353). We also compared the time scale of our study with that of other studies that found important effects of experimental duration. The larger time scale of our meta-analysis (~24 years) compared to other grassland (~15 years; Reich et al., 2012) and forest (~8 years; Huang et al., 2018) studies further justified our explanations (Lines 232-234).

Huang Y, et al. Impacts of species richness on productivity in a large-scale subtropical forest experiment. *Science* 362, 80-83 (2018).

Reich PB, et al. Impacts of biodiversity loss escalate through time as redundancy fades. *Science* 336, 589-592 (2012).

32. L 200-202. I'm not sure exactly why this is a shortcoming. Can you please explain before justifying?

AR: This is because several biodiversity experiments often refer to invaders as any species that has not been planted in a given experimental unit. Due to the coevolutionary history of species that are residents in other experimental units (i.e. internal invaders), they should be more likely to invade than novel external invaders —particular alien ones— that are not part of the experiment's resident species pool. This is also the reason why we distinguish between invader types. We added this information in the revised introduction (Lines 105-110).

33. L 214-216. This kind of statement is where you need to acknowledge the appropriate scope of the paper. You looked across grasslands, yet you say “clear evidence that plant diversity consistently increased invasion resistance of the resident community, mainly through selection effects, across various environmental conditions”. It may be as simple as stating that there is “clear evidence that in grassland systems,…”

AR: Revised as suggested (Lines 284-286). We also emphasized throughout the manuscript that our study is explicitly restricted to grasslands.

34. L 220-221. What “mechanistic insights”? Maybe instead of this sweeping statement give some concrete examples.

AR: We removed this sweeping statement from the conclusion.

35. L 256. 25 studies seems exceedingly low. Can you please justify this? What was the main bottleneck in removing studies?

AR: While there are numerous studies that have tested the biodiversity effect on invasion resistance, the bottleneck is that only few of them experimentally manipulated ECFs. We now mention this (Lines 275-277). We added the small number of experiments as a limitation of our study and call for more such experiments in the future (Lines 273-279).

36. L 263. This is too late to report this within methods, and should be in the main text as well.

AR: In the revised manuscript, we defined invasion in the abstract (Lines 21-22) and introduction (Line 58).

37. L 271. Can you test this both ways? Why would biomass v cover be preferable? This should be justified/explained.

AR: We prioritized the use of biomass data because 68% (17/25) of the experiments only reported biomass data and 16% (4/25) of the experiments only reported cover data. Only 16% (4/25) of the experiments provided both biomass and cover data.

In the revised manuscript, we added this information in the methods (Lines 343-345). We also found that excluding the four studies that only reported cover data did not affect our conclusions (Figs. S14–S16).

38. L 277. Did this include again both cover and biomass? Does mixing these impact results?

AR: Indeed, resident productivity included both cover and biomass data. We added this information (Line 351). Excluding the four studies that only reported cover data did not affect our conclusions (Figs. S14–S16).

39. L 283. This basically tells us the effect size between one and “more” species, but this “more” can differ drastically across studies, meaning that the NBE value can mean very different things in different studies. Can you expand on the range of “more”/”mix” species and potentially explicitly test for the effect of mix# species on the results of NBE? *Looks like you included this later.

AR: We presented the result of the effect of species number (i.e. the range of ‘more’/‘mix’ species) in Fig. 3.

40. L 296. Can you confirm that these values NBE values compared were calculated from the exact same # of mixture species?

AR: In the revised manuscript, we confirmed that Δ NBE was calculated by comparing ambient and manipulated environmental conditions pairwise for each diversity level of the resident community (Line 371).

41. L 315- 318. Looks like you included species richness here. However, which species richness you are referring to is unclear. Is this native richness?

AR: Indeed, it referred to native species richness in the mixtures. We explicitly mentioned this in the revision (Line 352).

REVIEWERS' COMMENTS

Reviewer #1 (Remarks to the Author):

Overall, the authors did a very good job in revising the manuscript and they have addressed all my initial comments. The introduction and discussion are much clearer now and also includes more conceptual background. I also appreciate the additional analysis that the authors did and that they provide in the supplementary.

I only have a few minor comments below that would improve the manuscript in my view.

L 66: I am not aware of any ecosystem that is not exposed to numerous ECFs, please remove "Most of"

L 108: I am not completely convinced about this argument as it is expressed right now. I think the authors could be more nuanced here and explain why they expect this pattern. I agree that co-evolutionary history can affect invasion success. However, shared co-evolutionary history has been shown to promote coexistence among (resident) plant species (see e.g. Thorpe et al. 2011, J of Ecology). Alien plants that are "new" to the community and thus share no co-evolutionary history with resident competitors, herbivores, soil organisms could also have an advantage over "resident invaders".

Thorpe, A. S., Aschehoug, E. T., Atwater, D. Z., & Callaway, R. M. (2011). Interactions among plants and evolution. *Journal of Ecology*, 99(3), 729-740.

L 246 More specific "to prevent the invasion by alien species"

Manuscript number: NCOMMS-24-01287A

Title: Biodiversity increases resistance of grasslands against plant invasions under multiple environmental changes

Authors: Cai Cheng, Zekang Liu, Wei Song, Xue Chen, Zhijie Zhang, Bo Li, Mark van Kleunen, Jihua Wu

Responses to reviewers' comments, Blue text = authors' reply, line numbers in blue color correspond to the revised version of the manuscript.

Responses to Reviewer #1

1. Overall, the authors did a very good job in revising the manuscript and they have addressed all my initial comments. The introduction and discussion are much clearer now and also includes more conceptual background. I also appreciate the additional analysis that the authors did and that they provide in the supplementary. I only have a few minor comments below that would improve the manuscript in my view.

Authors' response (AR): We appreciate the reviewer's positive evaluation and valuable comments on our revised manuscript. We revised the manuscript according to each of those comments. Please, see our point-by-point responses below.

2. L 66: I am not aware of any ecosystem that is not exposed to numerous ECFs, please remove "Most of".

AR: We removed "Most of" in the revised manuscript.

3. L 108: I am not completely convinced about this argument as it is expressed right now. I think the authors could be more nuanced here and explain why they expect this pattern. I agree that co-evolutionary history can affect invasion success. However, shared co-evolutionary history has been shown to promote coexistence among (resident) plant species (see e.g. Thorpe et al. 2011, J of Ecology). Alien plants that are "new" to the community and thus share no co-evolutionary history with resident competitors, herbivores, soil organisms could also have an advantage over "resident invaders".

Thorpe, A. S., Aschehoug, E. T., Atwater, D. Z., & Callaway, R. M. (2011). Interactions among plants and evolution. *Journal of Ecology*, 99(3), 729-740.

AR: We thank the reviewer for pointing this out. Because internal invaders are part of the experiment's resident species pool and have already existed in neighboring experimental units, they can be more likely to invade due to a priority effect. Therefore, we revised this sentence as "Among these invaders, species that are residents in other experimental units (i.e. internal invaders) should, due to a priority effect, be more likely to invade than novel external invaders —particular alien ones— that are not part of the

experiment's resident species pool." (Lines 98-101).

4. L 246 More specific "to prevent the invasion by alien species".

AR: We revised the sentence according to this suggestion (Line 239).